

# High throughput virtual screening of 230 billion molecular solar heat battery candidates

Mads Koerstz[1], Anders S. Christensen[2], Kurt V. Mikkelsen[1], Mogens Brøndsted Nielsen[1] and Jan H. Jensen[1]

[1] Department of Chemistry, University of Copenhagen, Copenhagen, Danmark, Denmark
[2] Department of Chemistry, University of Basel, Basel, Switzerland

## ABSTRACT

The dihydroazulene/vinylheptafulvene (DHA/VHF) thermocouple is a promising candidate for thermal heat batteries that absorb and store solar energy as chemical energy without the need for insulation. However, in order to be viable the energy storage capacity and lifetime of the high energy form (i.e., the free energy barrier to the back reaction) of the canonical parent compound must be increased significantly to be of practical use. We use semiempirical quantum chemical methods, machine learning, and density functional theory to virtually screen over 230 billion substituted DHA molecules to identify promising candidates. We identify a molecule with a predicted energy density of 0.38 kJ/g, which is significantly larger than the 0.14 kJ/g computed for the parent compound. The free energy barrier to the back reaction is 11 kJ/mol higher than the parent compound, which should correspond to a half-life of about 10 days—4 months. This is considerably longer than the 3–39 h (depending on solvent) observed for the parent compound and sufficiently long for many practical applications. Our paper makes two main important contributions: (1) a novel and generally applicable methodological approach that makes screening of huge libraries for properties involving chemical reactivity with modest computational resources, and (2) a clear demonstration that the storage capacity of the DHA/VHF thermocouple cannot be increased to >0.5 kJ/g by combining simple substituents.

## INTRODUCTION

The Sun is the most abundant source of energy, but periods of supply do not always match periods of demand. Therefore finding solutions for storing solar energy is one of the major challenges for a sustainable society. One approach is to employ light-induced isomerization of photoactive molecules (*Moth-Poulsen, 2013*; *Nielsen et al., 2020*) as exemplified by the dihydroazulene/vinylheptafulvene (DHA/VHF) thermocouple in Fig. 1. Upon irradiation, a molecule is converted to a high-energy photo-isomer and upon a certain stimulus, the high-energy isomer returns to the original molecule, and the excess energy is released as heat. This corresponds to a closed energy cycle of light-harvesting,

Corresponding author
Jan H. Jensen, jhjensen@chem.ku.dk

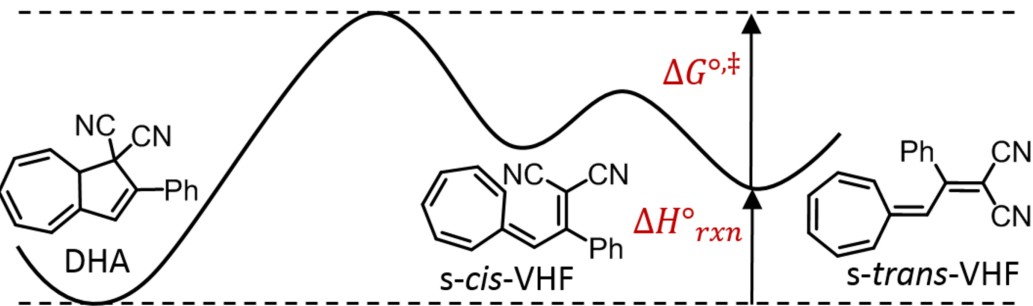

**Figure 1 Schematic representation of the dihydroazulene/vinylheptafulvene (DHA/VHF) thermal heat battery.** DHA is converted to VHF photochemically and the excess chemical energy ($\Delta H^{\circ}_{rxn}$) is released as heat when needed. The half life of VHF is determined by the free energy barrier of the back reaction to DHA. The molecule shown has been studied experimentally and is referred to as the "parent" molecule.

energy storage and release, with no emission of $CO_2$ or other chemical products. Such systems are termed molecular solar-heat batteries.

A suitable molecule (1) must absorb sunlight by converting to a higher energy form, (2) must absorb as much energy as possible (where ca 1 kJ/g is considered the practical maximum limit (*Kucharski et al., 2011*), but 0.3 kJ/g has been considered a reasonable target for some applications (*Bren et al., 1991*)) and (3) must be stable in the high-energy form for days or weeks. Dihydroazulenes (DHAs) are one class of promising candidates for solar heat batteries. The parent system (shown in Fig. 1) absorbs at the right wavelength with good quantum yield. However, the energy density is only 0.14 kJ/g and the half-life of VHF is only 3–39 h (depending on solvent) (*Broman et al., 2011*). Removal of one cyano group increases the storage density to 0.25 kJ/g and the half-life to years (*Cacciarini et al., 2015*). Unfortunately, the back reaction could not be triggered without causing degradation. With both cyano groups present the storage density can be increased by up to 0.38 kJ/g but not without decreasing the half-life significantly (*Skov et al., 2016*). The goal of this study is to identify substituted DHAs with both higher energy density and longer half life through high throughput virtually screening.

We consider 42 common substituents (including H) at seven different positions, as shown in Fig. 2, resulting in more than 230 billion molecules (slightly less than $42^7$ due to permutational symmetry). To our knowledge this compound library is the largest library considered thus far. For example, it is three orders of magnitude larger than the "ultra-large" library of 170 million compounds used by Shoichet, Irwin and coworkers (*Lyu et al., 2019*). While there has been a few virtual screening studies of thousands of reaction energies, (*Jinich et al., 2019*) corresponding screening studies of barrier heights typically involve less than 100 molecules (*Gani & Kulik, 2018*). Thus, screening barrier heights for billions or even thousands of molecules represents a fundamental challenge.

This paper is organized as follows. First we demonstrate the utility of semiempirical quantum mechanics (SQM) by screening all 35,588 singly and doubly substituted DHAs. Then we use a simple linear regression model and a gradient boosting decision tree trained on SQM data, to screen all 230 billion molecules (a flowchart of the exhaustive

| EWG | EDG |
|---|---|
| -[F, Cl, Br] | -OH |
| -CF$_3$ | -OMe |
| -CN | -NH$_2$ |
| -NO$_2$ | -NMe$_2$ |
| -CHO | -Me |
| -CO$_2$H | -NHC(O)Me |
| -C(O)Me | -SMe |
| -C(O)NH$_2$ | |
| -CCH | |
| -SO$_2$Me | |
| -CH=NH | |

**Figure 2 The substituents and positions considered in this study.** Substitutions at position eight is not considered, since removal of one cyano group has been shown to cause problems with the back reaction. The substituents are separated into electron withdrawing groups (EWG) and electron donating groups (EDG). There are a total of 42 different substituents counting hydrogen and phenyl resulting in more than 230 billion molecules (slightly less than 42$^7$ due to permutational symmetry).

screening procedure is shown Fig. S1). Finally, the linear regression model is then used to demonstrate that the best molecules predicted by that model can be found efficiently using a genetic algorithm (GA), suggesting that a GA could be used to screen even larger chemical spaces, perhaps using SQM directly rather than machine learning.

## RESULTS AND DISCUSSION

The goal of this study is to identify molecules with an energy storage density that is as high as possible and a half-life of that is at least as long as the parent compound (shown in Fig. 1) and preferably longer. We note that, according to transition state theory, even a modest 10 kJ/mol increase in the activation free energy corresponds to a 56 fold increase in the half life, which is 3–39 h for the parent compound (depending on solvent). Even high level ab initio calculations can easily give an error of ±10 kJ/mol for barrier heights, so molecules with large storage densities but computed barrier heights similar to the parent compound are potentially worth testing experimentally.

### Screening all singly and doubly substituted molecules

We start by screening all 35,588 singly and doubly substituted DHAs because they are most synthetically accessible and can be screened without machine learning models using semiempirical electronic structure methods (SQM). Figure 3A shows a plot of the SQM barriers plotted vs the storage densities for 32,623 singly and doubly substituted DHA/VHF couples. We want to select roughly 100 of the most promising molecules for further study with M06-2X/6-31G(d). As discussed in Supplemental Information, PM3 tends to overestimate the barrier relative to the parent molecule (the horizontal red line) so the barrier should be significantly higher than that. Similarly, GFN2-xTB tends to

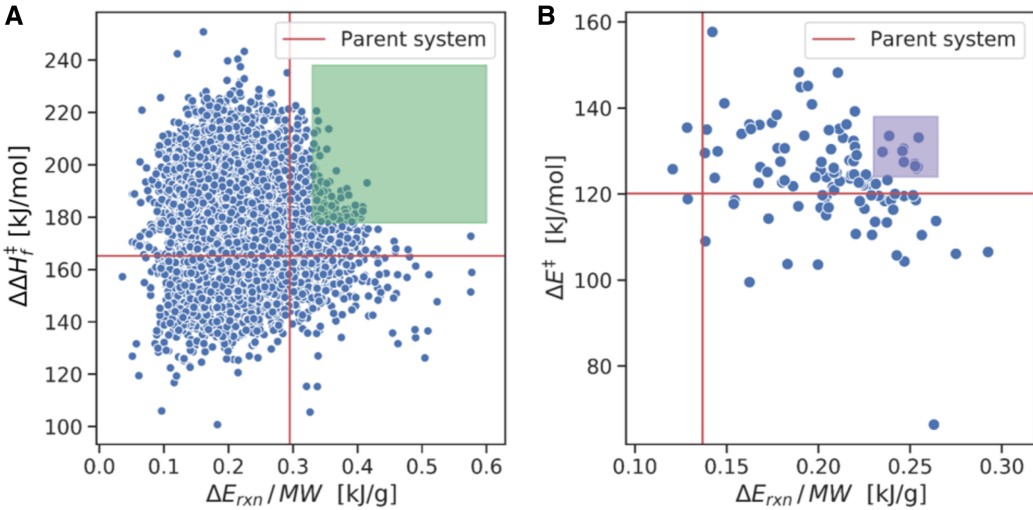

**Figure 3 (A) Electronic barrier heights and storage energies computed for 32,623 singly-and doubly substituted molecules using SQM and "5+5nrot" conformations as described in the "Computational Methodology" section. (B) Electronic barrier heights and storage energies computed for the 109 molecules in the green box in (A) computed using M06-2X/6-31G(d).** The geometry optimizations are started from the SQM structures and thepurple box highlights 10 molecules selected for further study (Table S1). The vertical and horizontal lines mark the storage energy and barrier height for the parent molecule illustrated in Fig. 1.

underestimate the reaction energy relative to the parent system (the vertical red line) and, hence, the energy storage density, so molecules with only somewhat higher storage density are potentially promising candidates. After some experimentation we found that cutoffs of 178 kJ/mol (15 kJ/mol higher than the parent compound) and 0.33 kJ/g leads to 109 molecules for further study using DFT (green box in Fig. 3A). For all 109 molecules we optimize the lowest energy GFN2-xTB DHA, VHF, and PM3 TS structures with M06-2X/6-31G(d) and the results are shown in Fig. 3B. As expected, the majority of the molecules have higher energy storage densities than the parent compound, but some have lower barriers with DFT. This indicates that by using PM3 energies, we are more likely to get false-positives than false-negatives. More false-positives will increase the number of redundant calculations; however, this is preferred over missing potentially excellent candidates.

From this set we choose ten molecules (the purple box in Fig. 3B; Table S1) for more thorough examination by performing a systematic conformational search (see "Computational Methodology" section) and computing enthalpies and free energies for the lowest energy conformers using M06-2X/6-31G(d).

The resulting DFT storage densities and free energy barrier heights are shown in Table 1 for the five molecules with largest storage density. The best five are all predicted to have an almost two-fold increase in storage density (0.24–0.25 kJ/g) compared to the parent system. Of the five, all but one have predicted back reaction-barriers that are between 4.7 and 16.5 kJ/mol higher than the parent compound.

Figure 3A shows three molecules with storage densities of nearly 0.6 kJ/g that we initially discounted because they are likely to have low barriers to the back reaction.

**Table 1 M06-2X/6-31G(d) predicted storage densities and back reaction barrier heights for the five molecules with largest storage densities among the molecules highlighted in Fig. 3B, based on the lowest free energy structures.** The corresponding M06-2X/6-31G(d)-values for the parent molecule are 0.14 kJ/g and 119.1 kJ/mol, respectively.

| Identifier | Structure | $\Delta H^{\circ}_{rxn}/MW$ [kJ/g] | $\Delta G^{\circ,\ddagger}$ [kJ/mol] |
|---|---|---|---|
| 1 |  | 0.25 | 112.8 |
| 2 |  | 0.25 | 135.6 |
| 3 |  | 0.25 | 123.8 |
| 4 |  | 0.24 | 124.2 |
| 5 |  | 0.24 | 129.9 |

To ensure that this is indeed the case, we perform the same systematic conformational search as for the ten promising molecules. The results are summarized in Table 2.

All three systems have an electron donating amino group at position 2 and an electron withdrawing group which allow for a hydrogen bond to the amino group in position 1. The three high energy density systems have very low back reaction barriers making them unsuitable for storage purposes. The amino group in position 2 have been shown to yield a large increase in storage energy, but also results in a significant decrease of the back reaction barrier Hansen et al. (2016). The hydrogen bond between the amino group and the electron withdrawing group stabilizes the DHA system increasing the storage energy, also locks VHF in the s-cis-VHF conformer. This means that the most stable VHF conformer is structurally very similar to the transition state structure, making the back reaction barrier very small.

To summarise, given the set of ligands shown in Fig. 2 the largest energy density one can hope to achieve with a singly or doubly substituted DHA molecules is about 0.5 kJ/g. But the half-life of the high energy states are much to short (sub-second) to be of practical

**Table 2** **M06-2X/6-31G(d) predicted storage densities and back reaction barrier heights for the three molecules with near 0.6 kJ/g storage density shown in Fig. 3A, based on the lowest free energy structures.** The corresponding M06-2X/6-31G(d)-values for the parent molecule are 0.14 kJ/g and 119.1 kJ/mol, respectively.

| Identifier | Structure | $\Delta H^{\circ}_{\mathrm{rxn}}/MW$ [kJ/g] | $\Delta G^{\circ,\ddagger}$ [kJ/mol] |
|---|---|---|---|
| 6 |  | 0.51 | 55.0 |
| 7 |  | 0.51 | 58.1 |
| 8 |  | 0.50 | 55.4 |

use. The largest possible energy density for molecules with half-lives on the order of hours or greater is around 0.25 kJ/g.

## Screening all 230 billion molecules

To exhaustively screen the entire chemical space a fast estimation of the storage density and back-reaction barrier is needed. We represent each molecule as a 287 (7 × 41) bit vector, where each row (7) represents one of the seven open positions on the DHA motif and each column (41) represents a non-hydrogen substituent. A more through description of the representation is shown in Fig. S2. Using this representation, we fit two linear regression models to SQM storage densities. One model predicts storage densities for molecules with ≤4 substituents and the other model for molecules with more than 4 substituents. We found that two different models are needed to get an acceptably low error (MAEs of 0.02 and 0.04 kJ/g, respectively). Each model is trained on ~25,000 molecules and tested on another ~25,000 molecules that represent a wide range of storage densities (Fig. S3). Figure S4 shows the number of times a ligand is found at a given position in the training set for the linear regression models. Though the distribution is somewhat uneven, all ligands are represented at all positions at least 40 times, so the training set is reasonably representative of the entire chemical space The RMSD values of the 1–4 substituent model and the 5–7 substituent model are 0.025 and 0.050 kJ/g, respectively (Fig. S5).

The simplicity of the linear ML-models means that they can be used to perform an exhaustive search of all 230 billion molecules in only 12 h using a single CPU. During the exhaustive search, all molecules with predicted storage densities smaller than 0.30 kJ/g are discarded (Fig. 4A). With the linear regression model, we reduce the chemical space of

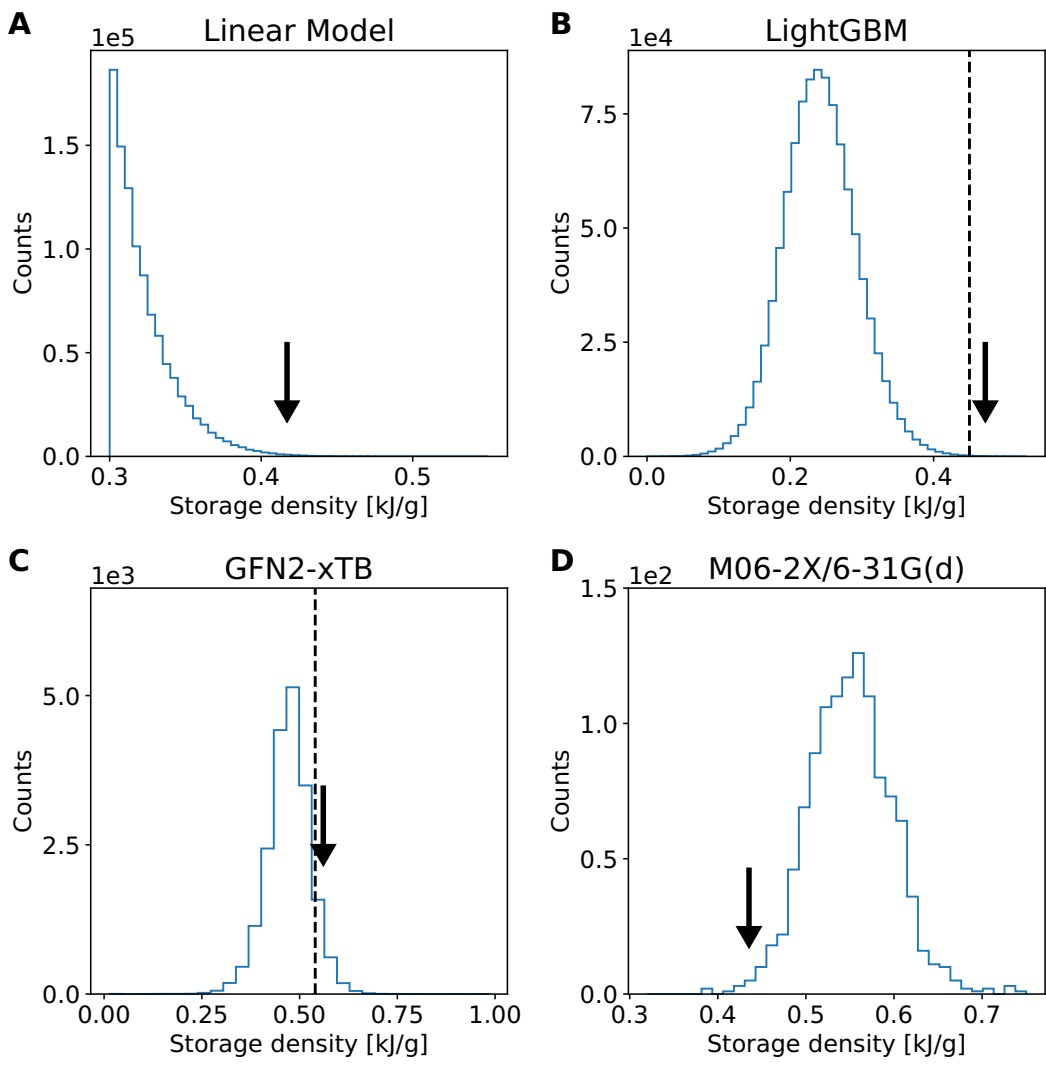

**Figure 4 Distribution of storage densities for (A) the linear regression model, (B) the LightGBM regression model, (C) GFN2-xTB and (D) DFT M06-2X/6-31G(d).** The black arrow indicates the storage density for 9 and the dashed line the cut-off used to select molecules for further study. The distribution of (A) and (B) does not show all 421 million molecules, but only a subset of 1 million randomly chosen molecules.               

interest from 230 billion to roughly 421 million molecules. As seen from Table S2, molecules with 5–7 substituents greatly outnumber molecules with 1–4 substituents.

The 421 million molecules are still far too many to screen using SQM methods, and the results from the linear model of 5–7 molecules indicate some degree of non-linearity (which are the group that the majority of the remaining molecules belong to). Thus, to more accurately predict the storage densities of the remaining molecules, we train a new model using the gradient boosted tree method, LightGBM (*Ke et al., 2017*). The LightGBM model is trained on new SQM data chosen among the remaining 421 million molecules. The new SQM data is chosen, such that the storage densities are evenly distributed between the minimum and maximum storage densities (0.30–0.57 kJ/g) as predicted by the linear regression model (Fig. S6). The LightGBM model is trained on

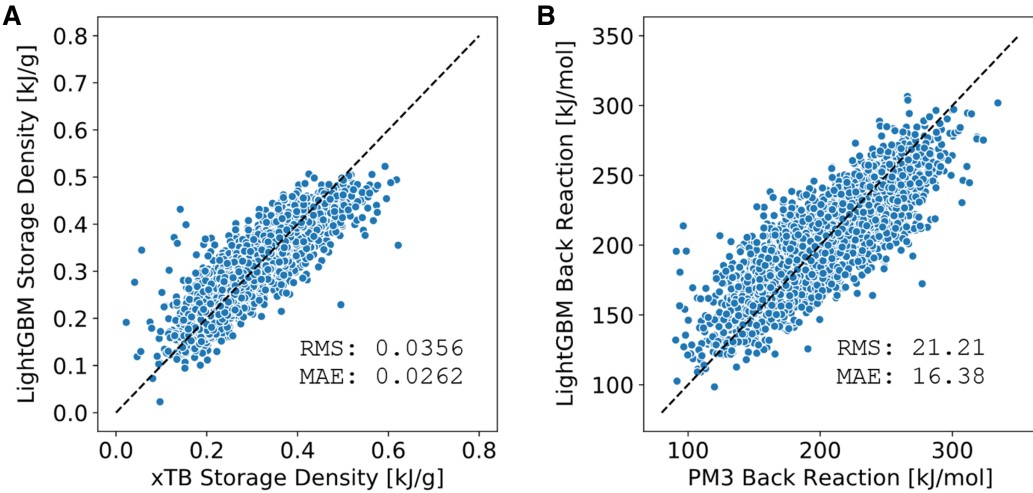

**Figure 5  Illustration of the performance of the LightGBM models for the test sets.** (A) The actual storage density computed using GFN2-xTB with 5+5nrot conformers compared to the predicted storage density using the LightGBM. (B) Comparison of the back barrier energy found using an PM3 adiabatic scan along the breaking bond to the predicted LightGBM model.

39,057 molecules and validated on 6,510 molecules, respectively. Finally, the new model is tested on 6,510 molecules and shown to have an RMSD value of 0.036 kJ/g (Fig. 5A), lower than the 0.050 kJ/g RMSD for the linear regression model for molecules with 5–7 substituents. The LightGBM model is then used to predict storage densities of all 421 million molecules in approximately 10 h.

Based on the storage densities predicted using the LightGBM model, we select 36,000 molecules and compute PM3 back reaction barrier heights to train and test a new LightGBM model. There is a rough inverse correlation between storage densities and barriers to the back reaction, so the molecules are selected in such a way that all storage density ranges are represented (Fig. S6). It proved challenging to converge all 36,000 transition state calculations, therefore the barrier heights are estimated using adiabatic scans along with the breaking bond. A LightGBM model was trained, validated, and tested using a 83/7/10 split of the data and yielded a RMSD of 21.21 kJ/mol (Fig. 5B). The model is used to predict the back reaction barriers of all the 421 million remaining molecules.

The LightGBM energies are then used to select molecules with a storage density of ≥0.40 kJ/g and a barrier height of ≥165 kJ/mol (the PM3 barrier for the parent system), of which there are 957,587. The highest storage density found in this set is 0.58 kJ/g so the main conclusion is that there are no molecules among the 230 billion molecules considered here with a storage density approaching 1 kJ/g. From these roughly 1 million molecules we need to select at most 50,000 molecules for SQM refinement. Figure 6 shows an overview of the number of molecules with various ranges of barrier heights and storage densities. After much deliberation we chose two sets for SQM refinement. The first set is the 1,969 molecules with barriers and storage densities of >165 kJ/mol and >0.50 kJ/g. Given the tendency of PM3 (and, hence, the LightGBM model) to overestimate the barrier height, these are almost certainly false positives but, given their high storage

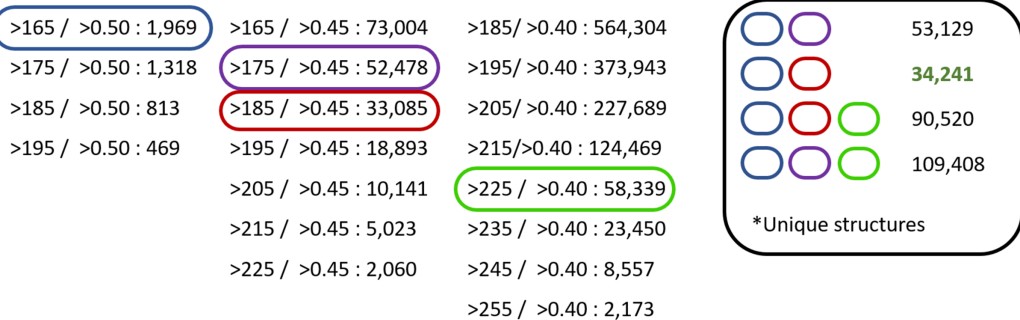

**Figure 6 Number of molecules with barriers and energy densities above certain thresholds.** The inset on the right shows the number of unique molecules for various combinations of subsets.

densities, we include them "just in case". The second set is the 33,085 molecules with barriers and storage densities of >185 kJ/mol and >0.45 kJ/g (Fig. 4B), which, given the higher barriers, are more likely to include true positives.

The combined sets of molecules contain 34,241 unique molecules, and out of these it was possible to compute the SQM storage density for 24,388 molecules. The remaining molecules did either not converge, or the connectivity changed during the optimization, and are therefore unlikely to be good candidates. For the remaining 22,258 molecules with storage densities above 0.4 kJ/g, we performed an adiabatic scan with PM3 to rapidly estimate the back reaction barrier height.

The next step is to select around 1,500 molecules for DFT calculations. Through experimenting with cut-offs, we pick 1,177 molecules with storage densities ≥0.54 kJ/g (Fig. 4C) and barriers ≥185 kJ/mol. The ground state SQM structures and transition state guess for the chosen molecules are then reoptimized with DFT. The optimizations succeed for 954 molecules, while the remaining 223 molecules fail due to the location of a wrong transition state (202 molecules) or because the connectivity of the product molecule changed during the optimization (21 molecules). We select random 20 molecules with wrong transition states and the 21 molecules with wrong products and perform a 15 point adiabatic scan along with the breaking bond in steps of 0.1 Å, staring from DHA. None of the 41 molecules showed any indication of a high-energy transition state by visual inspection of the scans, which indicates that the back reaction barrier is very low. Thus, the 223 molecules are not investigated any further.

Of the 954 molecules the majority have back reaction barriers that are smaller than the parent molecule. However, we do find one molecule, shown in Table 3, with a relatively high storage density and barrier height (Fig. 4D; Fig. S7). The molecule is subjected to a more thorough DFT investigation, as described in the supporting information, and the resulting reaction enthalpy and Gibbs free activation energy is shown in Table 3. The best thermal heat battery candidate is therefore **9** with a predicted energy density of 0.38 kJ/g and a barrier height of 130.0 kJ/mol. For comparison the energy density of the parent system (0.14 kJ/g) is almost three times lower, while the barrier height

**Table 3 M06-2X/6-31G(d) predicted storage densities and back reaction barrier heights for the two molecules, based on the lowest free energy structure.** The corresponding M06-2X/6-31G(d)-values for the parent molecule are 0.14 kJ/g and 119.1 kJ/mol, respectively.

| Identifier | Structure | $\Delta H^{\circ}_{rxn}$/MW [kJ/g] | $\Delta G^{\circ,\ddagger}$ [kJ/mol] |
|---|---|---|---|
| 9 | 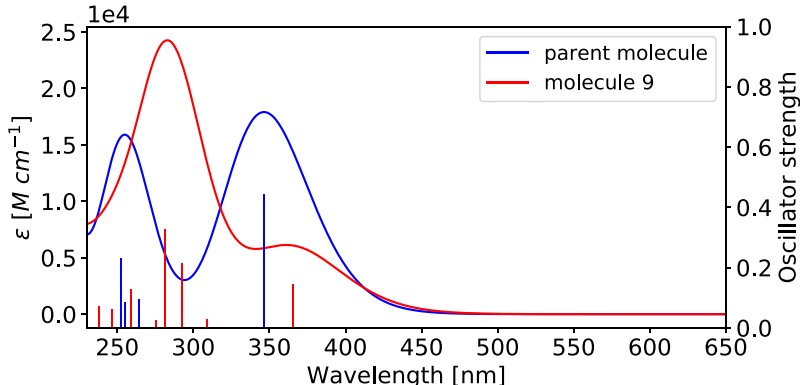 | 0.38 | 130.0 |

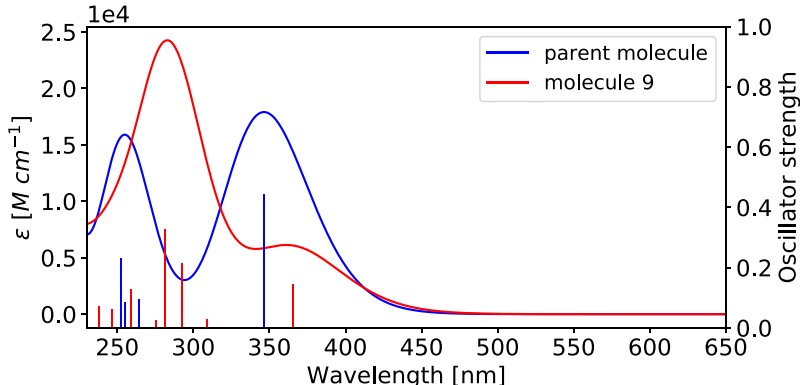

**Figure 7 Computed (CAM-B3LYP/6-311+G(d)//M06-2X/6-31G(d)) absorption spectra of compound nine and the parent compound.**

(119.1 kJ/mol) is 11 kJ/mol lower. According to TS theory, an 11 kJ/mol increase in barrier height increases the half life by a factor of 85, that is, from 3 to 39 h (depending on solvent) to about 10 days—4 months. The computed absorption spectrum of **9** is shown in Fig. 7, together with that of the parent system. Compound **9** retains an absorption peak abound 350 nm, but the intensity is reduced by about a factor of three.

## Test of GA search

While it is possible to exhaustively screen 42 substituents at seven positions, it will not be possible for future screening efforts that involve all nine positions and/or more than 42 substituents since the number of molecules will number in the trillions or more. We therefore test whether a GA can find the best candidate in such a large data set efficiently, using the linear regression model since we know what the correct answer is. To screen for both high storage density and long half life, we train a linear regression model to predict the back reaction barrier height using same molecules we used to train the storage density model (Fig. S8). Since the model is only used to test the efficiency of GA searches, we trained one model (instead of two as is done for the storage density).

   The best candidate in the search space is defined as the molecule with the largest storage density and a barrier height ≥180 kJ/mol. From the exhaustive search we know that this molecule is **10** shown in Table 4 with a storage density of 0.531 kJ/g.

**Table 4 Structure and rediscovery percentage for the molecule with largest storage density (0.531 kJ/g) and barrier height (190.37 kJ/mol) above 180.0 kJ/mol as predicted by the linear regression models.** The last columns gives the rediscovery percentage during 1,000 GA runs.

| Identifier | Structure | Rediscovery % |
|---|---|---|
| 10 |  | 74.0 |

For the GA each gene is defined as a vector with seven numbers (bases) ranging from 0 to 41, representing the seven possible positions and 42 possible substituents. The score for each gene is computed as

$$\text{score} = \min(\Delta E_{\text{rxn}}, t_{\text{rxn}})/t_{\text{rxn}} + \min(\Delta E^{\ddagger}, t_{\ddagger})/t_{\ddagger} \qquad (1)$$

where $t_{\ddagger}$ and $t_{\text{rxn}}$ are set to 180 kJ/mol and 1.0 kJ/g, respectively (*Brown et al., 2019*). The GA search starts with an initial population of 100 randomly generated genes of singly substituted molecules, for which the scores are computed using the linear regression models. Pairs of genes are then selected with a probability proportional to their score (roulette selection) and mated, by choosing a random cut point between bases for two parent genes and recombining. After each mating a mutation is performed 20% of the time by randomly changing one of the bases. This process is repeated 100 times and the 100 best scoring genes before and after mating is selected as the new population and the process is repeated for 100 generations.

We perform 1,000 separate GA searches and record the highest scoring molecule. **10** is found 74% of the time. Thus, one should be able to find the best molecule with 99% certainty by running only four GA searches, that is, by testing at most 40,000 different molecules. For comparison, we calculated storage densities and back-reaction barriers for >100,000 and >70,000 molecules as part of training and testing the machine learning models and their predictions. Thus, it may be computationally more efficient to search the chemical space using GA combined with SQM calculations rather than developing ML models for the properties of interest.

## COMPUTATIONAL METHODOLOGY

### Semiempirical calculations

$\Delta H^{\circ}_{\text{rxn}}$ is approximated as the difference in electronic energy ($\Delta E_{\text{rxn}}$) computed using GFN2-xTB (*Grimme, Bannwarth & Shushkov, 2017*) while $\Delta G^{\circ,\ddagger}$ is approximated as the difference in heat of formation ($\Delta\Delta H^{\ddagger}_{f}$) computed using PM3 (*Stewart, 1989*) (collectively referred to as "SQM"). GFN2-xTB is chosen due to its computational efficiency, while PM3 is chosen because it is available in both ORCA 4 (*Neese, 2018*) and Gaussian09 (*Frisch et al., 2009*) (see below). For each DHA structure the VHF structure is automatically generated using RDKit. $5 + 5n_{\text{rot}}$ random conformations (where $n_{\text{rot}}$ is the number of rotateable bonds in the molecule, see Fig. S9 for benchmark) are
generated using RDKit and optimized using GFN2-xTB (which are sufficient for screening purposes, illustrated in Fig. S10). Optimizations that result in discrepancies between the input and output connectivity are discarded. The lowest energy conformers of DHA and VHF are used to compute $\Delta E_{rxn}$. In some cases the conformational search is done systematically (see "DFT Refinement") at the SQM as described below.

To compute the energy barrier an adiabatic scan is performed (using ORCA) for the breaking CC bond, which is constrained to 12 values out to 3.5 Å starting from the DHA structure with the lowest energy. When screening all 230 billion molecules the highest energy structure and energy is used as an estimate for the transition state. However, for the single and double substituted case the highest energy structure is used as starting point for a transition state (TS) search using PM3 in Gaussian09 while computing the Hessian in each step. For the molecules where the TS search converged it is verified that the normal mode associated with the imaginary frequency lies along the reacting CC-bond. The low energy VHF GFN2-xTB conformer is reoptimized using PM3 and the PM3 barrier is computed. This TS connects DHA with with s-*cis*-VHF while the lowest energy VHF conformer is usually s-*trans*-VHF, so the implicit assumption is that s-*cis*-VHF and s-*trans*-VHF are in thermal equilibrium, that is, that the barrier between the two VHF conformations is lower than the barrier between s-*cis*-VHF and DHA (Fig. 1).

Figure S11 shows a comparison of barriers computed by locating the TS to barriers estimated by scans. There is a good correlation between the two barriers for medium sized barriers, but the scan tends to overestimate high barriers and underestimate low barriers. The use of scans to estimate barriers is this likely to lead to false positives, which are subsequently eliminated by DFT calculations, but is unlikely to lead to false negatives (i.e., we are unlikely to miss any promising candidates by using estimated barriers).

## DFT refinement

Select structures are investigated further at the M06-2X (*Zhao & Truhlar, 2008*)/6-31G(d) (*Hehre, Ditchfield & Pople, 1972*) level of theory (using Gaussian09). This level of theory was chosen as a good compromise between computational efficiency and accuracy as judged by comparison to DLPNO-CCSD(T) and CCSD(T)-F12a calculations (*Koerstz, Elm & Mikkelsen, 2017*). DFT is used in one of three ways. The first quick approach estimates DFT energies by reoptimizing the structure with the lowest SQM energy for DHA, VHF, and the TS and the electronic energy is used to compute the storage density and barrier. A benchmark investigation has been carried out by *Hansen et al. (2016)*.

The second approach is a more thorough DFT searches, used to obtain the final DFT energies. In the second approach a systematic conformer search is performed using SQM where each rotateable bond is rotated by ±120° starting from the lowest energy structure found using the RDKit conformer generating algorithm. Each structure is energy-minimized and in the case of the TSs the reacting bond is constrained. The minimized TS structure is then used as an initial guess for an unconstrained TS search. All unique conformers (conformers with Torsion Fingerprint Deviation *Schulz-Gasch et al. (2012)* that are less than 0.001 are considered identical) are then reoptimized with

M06-2X/6-31G(d) and the structures with the lowest free energy are used to compute the storage density and barrier.

## Machine learning models

We use both linear regression and a Gradient Boosted tree method (LightGBM) in this study. For the linear regression, a molecule is represented by positional seven-hot encoding, that is, vector with 287 (41 × 7) binary elements, where each chunk of 41 is one-hot encoded representation of a non-H substituent at a particular position (Fig. S2). This representation was used to train three different machine learning (ML)-models using SQM data. Linear regression and kernel ridge regression as implemented in Scikit-learn and a gradient boosted tree method as implemented in LightGBM. We found very little difference in performance between regression and kernel ridge regression and use the former, simpler, model in this study.

The combined use of positional binary encoding ($\mathbf{X}$) and linear regression amounts to an additive model

$$\Delta E = \mathbf{w} \cdot \mathbf{X} + b = \sum_{i=1}^{7} w_{ij} + b \tag{2}$$

where the regression coefficient $w_{ij}$ is the effect of placing substituent $j$ on position $i$ on either the reaction energy or barrier height and $b$ is the corresponding value for the unsubstituted molecule.

## SUMMARY AND OUTLOOK

We virtually screen 42 different substituents (including hydrogen) and seven possible substituent positions (Fig. 2) of the DHA/VHF thermal heat battery (Fig. 1) for molecules with high storage density ($\Delta H^{\circ}_{\mathrm{rxn}}$/MW) and stability ($\Delta G^{\circ,\ddagger}$) The size of the chemical space is roughly 230 billion molecules. We start by screening all 35,588 singly and doubly substituted DHAs using semiempirical methods (SQM): GFN2-xTB for the storage density and PM3 for the barrier height of the back reaction. Compared to M06-2X/6-31G(d), PM3 tends to significantly overestimate the barrier relative to the reference compound, while GFN2-xTB tends somewhat underestimate the storage energy, but the methods are sufficiently accurate to identify promising molecules for further refinement (Fig. S10).

The storage density and back reaction barrier of all 35,588 singly- and doubly-substituted DHA molecules are evaluated using SQM and used to identify 109 molecules for further study using M06-2X/6-31G(d) (Fig. 3A). The energy densities and barrier heights computed by reoptimising the lowest energy SQM-conformations are then used to select 10 molecules for further study using a thorough conformational search (Fig. 3B; Table S1). Five of the molecules are predicted to have an almost two-fold increase in storage density (0.24–0.25 kJ/g) compared to the parent system and all but one of these have predicted back reaction-barriers that are between 4.7 and 16.5 kJ/mol higher than the parent compound.

In order to screen the entire chemical space we generate additional SQM-data for higher degrees of substitution and use it to fit linear regression models that reproduce the storage

energies to within 0.017 and 0.038 kJ/g depending on degree of substitution (Fig. S5). These models are then used to estimate the storage density of all 230 billion molecules and the 421 million molecules with storage densities higher than 0.30 kJ/g are selected for further study (Fig. 4A). Gradient boosted tree method (LightGBM) models for the storage density and back reaction barrier are trained on new SQM data chosen among the these 421 million molecules, with respective MAEs of 0.026 kJ/g and 16.4 kJ/mol. These models are used to predict the storage densities and barrier heights for all 421 million molecules, and 34,241 molecules with storage densities larger than 0.45 kJ/g and high barrier heights are chosen for further study (Figs. 4B and 6). The highest storage density found in this set is 0.58 kJ/g so it is already clear at this point that there are no molecules among the 230 billion molecules considered here with a storage density approaching 1 kJ/g. The storage densities of the 34,241 molecules are computed using SQM and the barrier heights for the 22,258 molecules with SQM storage densities above 0.4 kJ/g are estimates using PM3.

Next, 1,177 molecules with storage densities ≥0.54 kJ/g (Fig. 4) and barriers ≥185 kJ/mol are selected for further refinement using DFT. The lowest energy geometry of DHA and VHF is reoptimized using M06-2X/6-31G(d), which also used to estimate the barrier height using adiabatic scans. Of the 954 molecules for which the DFT calculations succeed, the majority have back reaction barriers that are smaller than the parent molecule. However, one molecule, shown in Table 3, has a relatively high storage density and barrier height (Fig. 4D; Fig. S7) and is subjected to a more thorough DFT investigation.

Our conclusion is that the best thermal heat battery candidate among the 230 billion is **9** (Table 3) with a predicted energy density of 0.38 kJ/g and a barrier height of 130.0 kJ/mol. For comparison the energy density of the parent system (0.14 kJ/g) is almost three times lower, while the barrier height (119.1 kJ/mol) is 11 kJ/mol lower. According to TS theory, an 11 kJ/mol increase in barrier height increases the half life by a factor of 85, that is, from 3 to 39 h (depending on solvent) to about 10 days—4 months. The computed absorption spectrum of **9** is shown in Fig. 7, together with that of the parent system. Compound **9** retains an absorption peak abound 350 nm, but the intensity is reduced by about a factor of three.

The main conclusion of our work is that it is unlikely that the storage density of DHA can be increased to a value above 0.5 kJ/g by substitution at positions 1–7 (at least without decreasing the barrier to back reaction). Yet, storage densities above 0.3 kJ/g of both molecules **9** and **10** may be sufficient for some applications when considering also their long storage times. There are, however, other drawbacks with these molecules. For example, their absorption profiles are not optimum since there is reduced absorption in the visible region. And maybe more problematic, these molecules have rather complicated substitution patterns. There are today synthetic protocols available for functionalizing selectively at positions 1, 2, 3 and 7 (and in part position 6) of DHA, (Nielsen et al., 2020) but so far these protocols have only been used to introduce substituents efficiently at maximum three positions (see SI for further synthetic considerations). So new synthetic protocols need to be developed for introducing

substituents around the entire DHA core, and some suitable protecting groups need to be installed, for example to allow both amino and aldehyde functionalities in the same molecule. To avoid intermolecular reactions between such groups, it could be attractive to simply make **9** and **10** part of the monomeric repeat units of a polymer scaffold via the amine functionality. Organization of DHA units along a polymer may in fact possibly also enhance the energy density further as observed for some azobenzene-based materials (*Wu & Butt, 2019*). A Other design strategies will also be pursued in future work, such as replacement of the cyano groups at position 1 or the introduction of non-carbon atoms in the DHA scaffold (heterocyclic structures).

To our knowledge this compound library is the largest library considered thus far and the first to include barrier heights as a screening parameter. Notably, we show that the substituent effects on barrier heights can be estimated using ML and a very simple representation with sufficient accuracy to be useful. Despite the huge library size the screen is carried out using comparatively modest computational resources by using SQM as an intermediate step between ML and DFT calculations. SQM allows for large data sets for ML models to be constructed in a matter of days and the pre-screening tens of thousands of candidates so that DFT calculations are only performed on the most promising candidates. This is important because the prediction of reliable reaction energies and barrier heights requires thorough conformational searches that require significant computational resources at the DFT level. The overall methodological approach outlined in this paper is, in principle, generally applicable to lead optimisation of properties involving chemical reactivity. However, the accuracy of the SQM predictions relative to DFT and the accuracy of the ML models must of course be tested for each new system and not be adequate in all cases.

### Funding
This work was supported by a research grant (00022896) from VILLUM FONDEN. Anders S. Christensen received support from The National Centre of Competence in Research (NCCR) Materials Revolution: Computational Design and Discovery of Novel Materials (MARVEL) of the Swiss National Science Foundation (SNSF). The funders had no role in study design, data collection and analysis, decision to publish, or preparation of the manuscript.

### Grant Disclosures
The following grant information was disclosed by the authors:
VILLUM FONDEN: 00022896.
The National Centre of Competence in Research (NCCR).
Materials Revolution: Computational Design and Discovery of Novel Materials (MARVEL).
Swiss National Science Foundation (SNSF).

# PeerJ

## Competing Interests

Jan H. Jensen is an Academic Editor for PeerJ Physical Chemistry.

## Author Contributions

- Mads Koerstz conceived and designed the experiments, performed the experiments, analyzed the data, performed the computation work, prepared figures and/or tables, authored or reviewed drafts of the paper, and approved the final draft.
- Anders S. Christensen conceived and designed the experiments, authored or reviewed drafts of the paper, and approved the final draft.
- Kurt V. Mikkelsen conceived and designed the experiments, analyzed the data, authored or reviewed drafts of the paper, and approved the final draft.
- Mogens Brøndsted Nielsen conceived and designed the experiments, analyzed the data, authored or reviewed drafts of the paper, and approved the final draft.
- Jan H. Jensen conceived and designed the experiments, performed the experiments, analyzed the data, prepared figures and/or tables, authored or reviewed drafts of the paper, and approved the final draft.

## Data Availability

Code and data are available at GitHub (https://github.com/jensengroup/dha_htvs) and the University of Copenhagen (https://sid.erda.dk/sharelink/EwaEr2JMrb).

## Supplemental Information

Supplemental information for this article can be found online at http://dx.doi.org/10.7717/peerj-pchem.16#supplemental-information.

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
