# Peer review of "High throughput virtual screening of 230 billion molecular solar heat battery candidates"

_PeerJ Physical Chemistry, doi:10.7717/peerj-pchem.16_

## Round 0.1 · original submission · Minor Revisions

All our reviewers are very enthusiastic about your paper, and suggest several minor improvements.

·

Basic reporting

The manuscript is well-written and the background is well represented with appropriate citations.

1. The GitHub link is broken: https://github.com/jensengroup/dha should be https://github.com/jensengroup/dha_htvs

2. It is a bit confusing that compound 9 in Table 3 and compound 10 in Table 4 have storage energies and reaction barriers that are not comparable. As I understand it, the properties of 9 are predicted by the DFT calculations, while those of 10 are given by the linear model, but a quick look might lead to the belief that 10 is actually better than 9.

Experimental design

The research is scientifically sound and the research question well stated. All the code and modelling notebooks and data are made freely available online. The technical details are described in sufficient detail in the manuscript, and further details are available by inspection of the provided code. The QM methodology is very well benchmarked in the supporting information and it is obvious that a lot of work has gone into this manuscript. It combines many different aspects such as library enumeration, machine learning and automatic transition state searching with conformational search and the final product is very complete.

1. Regarding the machine learning, it is not clear to me what is the result of the train, validation and test sets. I would assume that the data in Figure 5, Figure S6 and Figure S8 is for the test set, but this should be clarified in the figures or the captions.

Validity of the findings

The presentation of the results and conclusions is very balanced, where the authors discuss also the limits of their methodology and the challenges associated with the synthesis of compound 9.

1. The molecules in Table 2 (6-7) are said to have low barriers for back reaction due to the hydrogen-bonding favoring the s-cis VHF form. Compound 10 has the same substitution pattern with an amine at position 2 and an aldehyde at position 1. Is compound 10 expected to have a similarly inflated reaction barrier as 6-7? In this case it should be highlighted as a limitation of the approach. This especially in light of the suggestion to replace the high-throughput virtual screening approach by the GA, which could get stuck in a false minimum.

2. The conclusion that “The overall methodological approach outlined in this paper is generally applicable to lead optimisation of properties involving chemical reactivity.” is too strong and should be qualified. Not all reactivity problems will be amenable to semi-empirical methods, particularly if challenging quantum-mechanically or if there are strong solvent effects. Also, it’s not certain that similar machine learning models employed will be equally successful to other reactivity problems.

4. The GA model finds compound 10 74% of the times, but it would be interesting to know the difference in predicted performance between 10 and the best molecule in the rest of the runs. If something nearly as good is found, that would increase the perceived usefulness of the GA even more.

Additional comments

I have read the report by Koerstz et al. with significant interest. They describe a computational high-throughput study for finding better molecular solar heat batteries of the photo-switch dihydroazulene/vinylheptafulvene. The goal is to find molecules with both a higher energy storage density and a higher energy for the back reaction (longer thermal half-life) than the parent compound. They first identify the most promising molecules with 1-2 substituents, for which it is possible to test all different combinations with quantum chemistry, using a combination of semi-empirical methods and DFT. They then go on to investigate molecules with 3-7 substituents, which is a too large set for QM calculations (ca. 230 billion molecules). Instead, they train machine learning models (linear regression and LightGBM) based on the semi-empirical results of a smaller set and use the predictions of the machine learning models to single out ca 30 000 candidates for SQM calculations. Of these, the ca 1000 most promising are subjected to more accurate but time-consuming DFT calculations. The DFT calculations identify a particular candidate, molecule 9, which has promising properties in terms of a higher barrier for back reaction and a greater energy storage density than the parent compound. However, it also has lower absorption in the visible region and is challenging to obtain synthetically. Finally, the authors demonstrate that a genetic algorithm can be used together with a linear regression model to assess both the energy storage and energy of back reaction. The GA is able to quite consistently recover (72% of the runs) the best molecule in the set predicted by the linear regression model and could therefore potentially serve as a more economical alternative to the high-throughput screening that led to the discovery of molecule 9.

As a suggestion for future work (does not need to be added to this paper!): For the NBD-QC photo switch, the energy barrier of the back reaction and the absorption spectrum are closely related in that an increased energy storage density leads to a lower barrier for back reaction (10.1021/acs.accounts.0c00235). It would be interesting to see if this trend broadly holds true also for the DHA/VHFs studied in this set and if so, if there are ways to break the relationship. Already the fact that 9 is predicted to absorb at a similar wavelength as the parent compound is an indication that the connection might be less strong. Potentially it could also be investigated with semi-empirical methods such as sTDA or INDO/S.

Reviewer 2 ·

Basic reporting

The authors performed a high-throughput virtual screening from a combinatorial library they generated from DHA to find a suitable thermocouple pair. This methodology used semiempirical methods, DFT calculations, and machine learning methods. They found the best candidate has an energy density of 0.38 kJ/g and a half-life of 10 days 4 hours.

The English through the text is clear and unambiguous. I consider that it has a well-provided context and pertinent references. The Figures are overall clear and well described in the text; with the exception of figure 3, the graph labels slightly differ from what is defined in the text. I consider that the manuscript is written in a professional way, its structure is clear, and has a concise methodology.

I consider that this manuscript meets the standard of the journal, its pertinence is well documented and the methodology is clear and unambiguous.

Experimental design

The authors defined the research question in a clear way; they establish the methodology to find the best possible candidate using a scaffold analysis, they first consider single and double substitutions on the DHA core to calculate energy barriers and storage energies using PM3 Hamiltonian and screening the 10 best molecules from there to be calculated using DFT M06-2X/6-31G(d) level of theory. From these results, the authors trained a linear regression model to screen the energy barriers and storage densities of the 42^7 compounds; then focusing on storage density values greater than 0.3 kJ/g, they reduced this library to 421 M molecules.

Following the results in this reduced library, the authors trained a gradient boosted model with new PM3 calculations and performed a binned sampling to get 957k molecules which energy barrier and storage density are above 165 kJ/mol and 0.4 kJ/g, respectively. They further performed a refinement to get 34241 unique molecules to be calculated using PM3 and found around 22 k molecules with storage density above 0.4 kJ/g and perform on these a PM3 adiabatic scan, giving 1177 suitable candidates that went under DFT refinement to find the best candidate having a storage density of 0.38 kJ/g and a barrier height of 130 kJ/mol.

This methodology design heavily relies on PM3 calculations, which are suitable for the organic molecules considered in this library. I consider that the ML methods used (linear regression and gradient boosting decision tree) are well implemented. The authors bound the capabilities of their approach and give the community a deep insight into the energetics and storage density of these parent couples. Furthermore, they applied a genetic algorithm and get comparable results to the screening process.

Validity of the findings

I consider that the findings of this work are well supported by the computational evidence the authors provided.

The methodology is clearly discussed. I consider that the conclusions are insightful and well supported by the evidence provided.

The authors perform a well-organized methodology to answer their research question.

They shared their datasets and code as public repo (github).

Additional comments

I would suggest that the authors please be more explicit in how is the process of vectorizing the molecules to be feed to the ML methods are. Or if they are using a standard algorithm such as Morgan fingerprints.

On synthetic accessibility, I’m aware that a quite small number of the possible combinatorial functionalizations could lead to synthetic viable molecules. I may suggest that, at least for the small screening library (the one tested using PM3 and TB methods), the authors could evaluate the synthetic complexity score (https://github.com/connorcoley/scscore) to have a complexity metric that can enrich the discussion of their results.

In Figure 3, the labels of electronic barriers and storage energies do not coincide as the same variables in the first paragraph of the Computational Methodology, please uniform the notation.

·

Basic reporting

- The computational pipeline introduced in this paper is very impressive and clearly well thought out and implemented. However, it is complex, with several steps involved, each using different subsets of data with different methods and models. Therefore perhaps a schematic diagram of the entire pipeline, either as a main or SI figure, would help guide the reader.
- In the manuscript, when the authors mention the number of substituents considered, they sometimes write “42” and other times “41” Please make this consistent.

Experimental design

no comment

Validity of the findings

- Can the authors add a comment somewhere in the main text on the possibilities or outlooks for comparing the predictions to experimental data? The project hinges on the assumption that the quantum chemistry calculations are an accurate representation of the ground truth values for storage energies and reaction barriers. So additional comments on how those DFT calculations are expected to correlate with experimental data would be valuable. (More specifically: I have read the section “Synthetic considerations”. However, what I’m referring to is the possibility of connecting the predicted thermodynamic/kinetic parameters to a database of experimental data.)
- In the summary and outlook, pg. 15, the authors mention “For example, their absorption profiles are not optimum.” Can they add an extra sentence/phrase to explain why?

Additional comments

- [OPTIONAL] One really nice part of the paper is when they select 109 molecules for further study with DFT. They demonstrate that the majority of the selected molecules have higher energy storage densities than the parent compound. While this is a very promising result, their argument/evidence would be strengthened even more if a statistical measure was attached to this finding. More specifically, what is the probability of finding a similar fraction of compounds with higher energy storage (than the parent compound) for random sets of 109 molecules (sampled several times). In other words, a p-value for the observed enrichment of high energy storage compounds. I’m not sure if this is computationally prohibitive, so that is why I’m tagging this as “optional”.
- In summary and outlook, pg. 14, third paragraph, there is a typo: “421 molecules” should be “421 million molecules”.
- Figure S6: The acronym “ML” appears in the figure legend, but the exact meaning of ML is not specified in the caption. Please fix this.
- Figure S2: Fig. S2 (A) has green data points that are circled in black. However, no mention of what this means appears in the figure legend. I know this is mentioned in the SI text, but adding this to the figure legend might help with clarity.
- I think it’s great that the authors provide access to the code through a repository.

---

## Round 0.2 · accepted · Accept

Thank you for addressing all the issues. I am glad to accept your manuscript.

·

Basic reporting

All my concerns have been adequately addressed.

Experimental design

All my concerns have been adequately addressed.

Validity of the findings

All my concerns have been adequately addressed.

Reviewer 2 ·

Basic reporting

no comment

Experimental design

no comment

Validity of the findings

no comment

Additional comments

Thanks for responding to my comments. I consider the manuscript is in good shape to be published in this journal.

·

Basic reporting

no comment

Experimental design

no comment

Validity of the findings

no comment

Additional comments

The authors have addressed my previous comments and I have no further suggestions for improvements. The article should be accepted as is.